# The Rho/Rac Guanine Nucleotide Exchange Factor Vav1 Regulates Hif-1α and Glut-1 Expression and Glucose Uptake in the Brain

**DOI:** 10.3390/ijms21041341

**Published:** 2020-02-17

**Authors:** Jaewoo Hong, Yurim Kim, Sudhirkumar Yanpallewar, P. Charles Lin

**Affiliations:** 1Cancer and Inflammation Program, Center for Cancer Research, National Cancer Institute, Frederick, MD 21702, USA; yoolim567@naver.com; 2Department of Biology, University of Maryland, College Park, MD 20742, USA; 3Mouse Cancer Genetics Program, Center for Cancer Research, National Cancer Institute, Frederick, MD 21702, USA; sudhirkumar.yanpallewar@nih.gov

**Keywords:** Vav1, HIF-1α, GLUT-1, learning deficiency, hypoxia

## Abstract

Vav1 is a Rho/Rac (Ras-related C3 botulinum toxin substrate) guanine nucleotide exchange factor expressed in hematopoietic and endothelial cells that are involved in a wide range of cellular functions. It is also stabilized under hypoxic conditions when it regulates the accumulation of the transcription factor HIF (Hypoxia Inducible Factor)-1α, which activates the transcription of target genes to orchestrate a cellular response to low oxygen. One of the genes induced by HIF-1α is GLUT (Glucose Transporter)-1, which is the major glucose transporter expressed in vessels that supply energy to the brain. Here, we identify a role for Vav1 in providing glucose to the brain. We found that Vav1 deficiency downregulates HIF-1α and GLUT-1 levels in endothelial cells, including blood-brain barrier cells. This downregulation of GLUT-1, in turn, reduced glucose uptake to endothelial cells both in vitro and in vivo, and reduced glucose levels in the brain. Furthermore, endothelial cell-specific Vav1 knock-out in mice, which caused glucose uptake deficiency, also led to a learning delay in fear conditioning experiments. Our results suggest that Vav1 promotes learning by activating HIF-1α and GLUT-1 and thereby distributing glucose to the brain. We further demonstrate the importance of glucose transport by endothelial cells in brain functioning and reveal a potential new axis for targeting GLUT-1 deficiency syndromes and other related brain diseases.

## 1. Introduction

Energy sources and oxygen are delivered to organs and tissues throughout an organism via the vascular system, which acts as a critical lifeline for cellular metabolism. For the delivery of these substrates to parenchymal cells, they must pass across the endothelium [1,2,3,4]. Glucose is one of the primary sources of cellular energy, and endothelium plays a vital role in tissue glucose metabolism. A key regulator of this endothelial metabolic function is the oxygen-sensing transcription factor, HIF-1α [5]. HIF-1α maintains glucose homeostasis and is a well-known regulator of GLUT-1, which is the primary transporter of glucose across the endothelial blood-brain barrier [6]. It has been reported that endothelial cells from HIF-1α knock-out mice showed decreased GLUT-1 levels, followed by reduced glucose uptake [5]

The endothelium regulates glucose metabolism in parenchymal cells via two major pathways: by transporting glucose from blood to tissue; and by presenting endothelial endocrine or paracrine factors to the parenchyma. The first pathway involves the uptake of primary glucose across the membrane, which occurs by facilitated diffusion in an energy-independent manner by glucose transporters [3]. Glucose transporters are a structurally homologous family of proteins that are expressed in a tissue-specific way [7,8]. Among the family members, GLUT-1 is the major glucose transporter in endothelial cells, including the blood-brain barrier [7,9,10,11]. Human GLUT-1 malfunctioning mutations are associated with a reduction of glucose uptake to the brain and the tendency of seizures [2,12]. GLUT-1 mutation is also a potential cause of CNS syndromes, including learning disabilities [9]. GLUT-1 is also expressed in glia but to a lesser extent than in endothelial cells [10]. Although the major glucose transporter of neurons is GLUT-3, defective regulation of the massive influx of glucose from endothelial cells to the parenchyma, including neurons, is likely to be an underestimated cause of neurological dysfunctions.

Vav proteins are a family of Rho/Rac guanine nucleotide exchange factors (GEF) that mediate a wide variety of cellular functions. They consist of three highly conserved members in vertebrates. Vav1 is restricted mainly to hematopoietic cells, where it acts as an essential signaling factor involved in hematopoietic cell activation, growth, and differentiation [13,14]. Mice lacking Vav1 are viable and fertile and generally develop except for a partial defect in lymphocyte development [15,16,17]. Vav1 is also expressed in endothelial cells where it activates Rac1 and Cdc42 in association with δ-catenin, which may be involved in regulating cell motility and vascular network formation [18]. In our recent study, Vav1 was identified as a critical regulator of the vascular response to hypoxia: hypoxia blocks Vav1 degradation, whereas it is continually produced and degraded under normoxic conditions. This accumulation of Vav1 leads to hypoxia-induced HIF-1α accumulation. Indeed, in mice lacking Vav1 that were subjected to cardiac ischemic stress, the absence of HIF-1α activation led to death [19].

The brain is directly affected by energy regulation [20,21]. It utilizes glucose at a high rate, and a lack of glucose has been linked to various brain-related diseases, including dementia [22,23,24]. The dysregulation of glucose metabolism in brain damage has been a focus of therapeutic strategies because of the possibility to administer soluble molecules that stimulate glucose metabolisms, such as insulin or glucagon-like peptide 1 (GLP-1). These molecules improve cognitive responses in humans diagnosed with Alzheimer’s disease, as well as in mouse models [25,26,27,28,29,30,31]. These reports suggest that the dysregulation of glucose in the brain is crucial in neurologic diseases, as well as the progression of other metabolic diseases, including diabetes.

Given that Vav1 increases the stability of HIF-1α and HIF-1α is the critical transcription factor for GLUT-1 [19], we hypothesized that Vav1 regulates the expression and function of GLUT-1 via HIF-1α, thereby regulating glucose metabolism in the brain. Glucose deficiency in the brain is directly related to various neurologic dysfunctions, including learning disabilities. In this study, we suggest the regulation of glucose distribution is Vav1 dependent, and Vav1 is an important factor for the brain function.

## 2. Results

### 2.1. Vav1 Regulates HIF-1α and GLUT-1 in Endothelial Cells

In order to verify whether Vav1 regulates HIF-1α in endothelial cells, we incubated human umbilical vein endothelial cells (HUVEC) after knocking down Vav1 in normoxic or 1% O2 hypoxic conditions for 5 h. HIF-1α was strongly upregulated in hypoxic conditions in control cells, whereas it remained largely undetectable in Vav1 knock-down cells (Figure 1A). Vav1 was also upregulated when cells were incubated in hypoxic conditions, consistent with our previous findings [19]. Since HIF-1α induces the transcription of GLUT-1, only 5-h of hypoxia did not show any increase in GLUT-1 expression, as expected [32,33]. Therefore, we incubated Vav1 knock-down HUVEC in hypoxic conditions for 16 h and measured GLUT-1 levels. Both HIF-1α and Vav1 were upregulated from control cells and HIF-1α was downregulated when Vav1 was knocked down in hypoxic condition. Interestingly, GLUT-1 levels increased compared to normoxic conditions, whereas there was no difference in the Vav1 deficient HUVEC (Figure 1B). Next, we performed a similar experiment in the human blood-brain barrier endothelial hCMEC/d3 cells to validate whether GLUT-1 is controlled by Vav1 and HIF-1α. As expected, there was no upregulation of HIF-1α or GLUT-1 under hypoxic conditions in the Vav1 knock-down cells (Figure 1C). We further verified that HIF-1α controls GLUT-1 at the mRNA rather than the protein level by overexpressing HIF-1α in HUVEC. We found that the mRNAs of HIF responsible element (HRE) genes, including GLUT-1 and VEGF, were upregulated by HIF-1α overexpression in both normoxic and hypoxic conditions (Appendix A). To further verify that GLUT-1 is controlled by Vav1, we overexpressed HIF-1α in control or Vav1 shRNA-introduced hCMEC/d3 cells and incubated in hypoxic condition for 5 h. Consistent with our previous results, Vav1 shRNA transfected cells showed downregulated HIF-1α, followed by GLUT-1 levels compared to control shRNA transfected cells (Figure 1D). However, overexpression of HIF-1α blocked this downregulation. This GLUT-1 rescue shows that Vav1 controls HIF-1α, which in turn regulates GLUT-1 transcription in endothelial cells.

### 2.2. Vav1 Regulates Glucose Uptake in Endothelial Cells

Next, we determined whether Vav1 affects the ability of GLUT-1 to take up glucose. Vav1 shRNA or control shRNA were introduced into HUVEC or hCMEC/d3 cells. We treated these cells with 2-Deoxy-2-[(7-nitro-2,1,3-benzoxadiazol-4-yl) amino]-D-glucose (2-NBDG), a fluorescent glucose analog under hypoxic conditions, after starvation in the glucose-free medium for 1 h. The Vav1-deficient groups of both HUVEC and hCMEC/d3 showed a decreased 2-NBDG signal (Figure 2A). To quantify this, we measured the mean fluorescent intensity of 2-NBDG from both HUVEC (Figure 2B) and hCMEC/d3 cells (Figure 2C). Compared to control cells, the Vav1 knock-down cells showed a significantly reduced 2-NBDG signal. This reduction was comparable to that seen upon treatment with the GLUT-1 inhibitor, apigenin. These results demonstrate that, under hypoxic conditions, Vav1 regulates glucose uptake through GLUT-1 in endothelial cells.

### 2.3. Vav1 Deficiency Delays Glucose Uptake In Vivo

We next determined whether Vav1 regulates glucose uptake also in vivo by testing glucose uptake from the bloodstream in endothelial-specific Vav1-deficient mice. VE-Cadherin-cre/ERT-Vav1flox/flox mice were treated with 4-OH tamoxifen for two weeks to generate endothelial-specific Vav1-deficient mice. First, we measured basal levels of blood glucose in wild type and Vav1-deficient mice. In the absence of endothelial Vav1, blood glucose levels were relatively upregulated, even without glucose administration (Figure 3A). Next, an intraperitoneal glucose tolerance test was performed to compare the uptake of administered glucose. Compared to the wild type group, Vav1-deficient mice showed significantly increased levels of blood glucose and delayed recovery back to basal levels (Figure 3B). Over 50% of the Vav1-deficient mice had basal blood glucose levels of 126 mg/dl or more, compared to less than 20% of the wild type mice (Figure 3C). Furthermore, Vav1-deficient animals were significantly heavier than wild type animals (Figure 3D). These characteristics are found in diabetic mice, although we could not detect any related histological structural changes in the tissues of Vav1-deficient mice in high energy-consuming organs and β-cell structure in pancreas (Appendix A).

### 2.4. Vav1 Deficiency Downregulates Glucose Distribution in the Brain

We focused on the brain of endothelial Vav1-deficient mice because the brain is one of the major organs that utilizes glucose. As expected, GLUT-1 was expressed in the brain of wild type mice, but, notably, was mostly undetectable in the endothelial Vav1-deficient mice (Figure 4A). ^18^F- Fludeoxyglucose (FDG) is widely used in positron emission tomography (PET) to measure inflammatory regions, especially in cancer patients. FDG is also an analog of glucose, which is taken up by glucose transporters. We compared the signal of injected FDG in wild type and endothelial Vav1-deficient mice using PET. The FDG signal was significantly lower in the brains of Vav1-deficient mice compared to wild type mice. The dilated heart also showed a high signal because it contains high amounts of blood, as well as the gall bladder and urinary bladder, which contain high levels of FDG metabolite. When the PET image was superimposed onto the CT image, it is clearer to observe FDG was less distributed in Vav1-deficient mouse brain (Figure 4B). Both the mean and total standardized uptake value (SUV) was significantly lower in Vav1-deficient mice (Figure 4C,D). Next, we evaluated the effect of endothelial Vav1-deficiency on the brain function by subjecting wild type and endothelial Vav1-deficient mice to a fear conditioning test (see Materials and Methods), which represent a form of learning [34]. We found a significant decrease in the percentage of time spent freezing to context, i.e., in the same environment as the conditioning in the endothelial Vav1-deficient mice compared to the wild type mice. This difference was lost when the mice were analyzed for freezing to cue, i.e., in a different environment but with the same stimulus (Figure 4E). This suggests that there is some impairment of learning when Vav1 is deficient. These results show that Vav1 in the endothelium regulates glucose distribution to the brain as well as the function of the brain, such as learning.

## 3. Discussion

Since GLUT-1 is a component of HIF Responsible Elements (HRE) and is highly expressed in endothelial cells, we focused on glucose uptake in endothelial cells. As we expected, the Vav1 knock-down led to the downregulation of both HIF-1α and GLUT-1. In our experiments, HIF-1α was upregulated by hypoxia in 5 h, but GLUT-1 was not increased. Since HIF-1α controls the transcription of GLUT-1, the upregulation of proteins was not shown at the same time in HUVEC. We could observe upregulated GLUT-1 after 16-h incubation of cells in hypoxia, unlike HIF-1α in HUVEC. Since HIF-1α is a transcription factor, GLUT-1 protein expression might be affected only in extended exposure in hypoxia.

Interestingly, hCMEC/d3 showed a slightly different pattern to HUVEC. hCMEC/d3 showed increased HIF-1α and GLUT-1 together after 16-h incubation in hypoxia. This difference may be dependent on cell type. Notably, the upregulation of HIF-1α and GLUT-1 was abrogated by Vav1 knock-down in two kinds of cells. Additionally, the GLUT-1 decrease was rescued when HIF-1α was overexpressed in Vav1 knock-down cells, which demonstrates that Vav1 is upstream of both HIF-1α and GLUT-1.

The basal blood glucose level of Vav1 knock out mice was significantly higher than that of wild type mice, which may be because of a lower uptake of glucose from the bloodstream to the endothelial cells. In the Vav1-deficient group, exogenously administered glucose was not taken up as quickly as in wild type mice. Additionally, the increase in glucose levels after injection was higher than in the wild type mice. This in vivo data indicates that glucose distribution in Vav1-knockout mice is delayed. Furthermore, more than 50% of Vav1-knockout mice had over 125 mg/dl of blood glucose, while less than 20% of wild type mice had these high levels. Vav1-knockout mice also showed increased body weight compared to wild type mice. High blood glucose and body weight are characteristics of diabetic mice. Other than this, the endothelial-specific Vav1-knockout mice were fertile and had no gross phenotypic abnormalities.

The brain requires a consistent supply of energy to maintain proper functioning, and it is the second most glucose-consuming organ after the liver. GLUT-1 is the main route to supply glucose from the bloodstream to the brain. Thus, loss of GLUT-1 in endothelial cells of the brain is directly linked to brain damage and related to several diseases, including GLUT-1 deficiency syndrome and Alzheimer’s disease [35,36]. Because Vav1 deficiency leads to the loss of GLUT-1, the distribution of glucose to the brain is remarkably decreased. In our studies, we observed that Vav1 deficiency not only downregulates glucose distribution to the brain but also memory deficiency observing the behavioral experiment. Considering these findings, we propose that endothelial Vav1 is a critical factor that controls brain function, including memory.

GLUT-1 deficiency syndrome is a genetic disorder that varies in symptoms and severity between individuals. For example, some individuals with GLUT-1 deficiency syndrome do not develop epilepsy. Other possible symptoms include abnormal eye-head movements, body movement disorders, developmental delays, and varying degrees of cognitive impairment, slurred speech, and language abnormalities [35]. Although Vav1 has not been previously associated with conditions caused by the mutation of SLC2A1, which encodes GLUT-1, the phenotype of endothelial Vav1-deficient mice was very close to GLUT-1 deficiency syndrome, supporting further studies such as discovering the pathological mechanism and in vivo experiments. Glucose uptake deficiency in the brain is not only related to GLUT-1 deficiency syndrome but also to Alzheimer’s disease associated with type 2 diabetes and various other brain disorders [18,22,23]. In light of our results, it will be interesting to study the potential role of Vav1 in these disorders.

It is well-known that HIF-1α is a crucial transcription factor for GLUT-1. Our previous study revealed that Vav1 is continuously produced and degraded under normoxic conditions, whereas under hypoxic conditions, it is stabilized and required for HIF-1α activation via activation of p38 MAPK [19]. We now take the next step and show that Vav1-mediated HIF1 α activation is required for glucose uptake in the brain.

## 4. Materials and Methods

### 4.1. Cell Culture

Human umbilical vein endothelial cells (HUVECs) and the hCMEC/d3 human blood-brain barrier cell line were obtained from Lonza (Walkersville, MD, USA) and Sigma (St. Louise, MO, USA), respectively. The cells were cultured according to the manufacturer’s protocols. For hypoxia, cells were incubated in 1% O_2_ adjusted incubator (Thermo, Waltham, MA, USA) for 5 h or 16 h.

### 4.2. Transfection and Lentiviral Transduction

Lentiviral control and Vav1 shRNA constructs were obtained from Sigma. Lentiviral HIF-1α construct was obtained from Addgene (Watertown, MA, USA). Constructs were prepared as lentivirus to transduce knockdown or overexpression. Briefly, lentiviral vectors were co-transfected with VSV.G and envelope vectors to 60% confluent 293T cells in 10-cm culture plates using Fugene HD (Promega, Madison, WI, USA). After three days of incubation, the culture supernatant was collected and concentrated to 500 μL using Lenti-X concentrator (Takara, Japan). A total of 8 μg/mL of polybrene and 10 μL of concentrated virus was treated with HUVEC or hCMEC/d3 and then incubated for 48 h for transduction.

### 4.3. Glucose Uptake Assay

Glucose uptake assay was performed with Glucose Uptake Cell-based Assay Kit (Cayman Chemical, Ann Arbor, MI, USA) following the manufacturer’s description. As part of this process, 2-NBDG and apigenin were diluted and treated following the manufacturer’s description (Cayman Chemical, Ann Arbor, MI, USA). Cells were starved in glucose- and serum-free RPMI medium for 1 h before assays. After treatment, cells were observed by LSM780 confocal microscopy (Zeiss, Germany). The mean fluorescence was measured directly from 96-well plates using a luminometer (Veritas, Sata Clara, CA, USA)

### 4.4. Western Blot

Cells were lysed directly with Laemmli sample buffer and briefly ultrasonicated. The lysate was boiled at 100 C for 5 min and then 1 μg from cell lysate or 5 μg of tissue lysate was subjected to SDS-PAGE and transferred to PVDF for western blot analysis. The blot was blocked for 1 h in 5% non-fat dry milk containing TBST. Primary antibodies against HIF-1α (#36169), GLUT-1 (#12939), and Vav1 (#4657) were purchased from Cell Signaling Technology (Danvers, MA, USA) and probed at 1:1,000 dilution in the blocking solution. Secondary antibodies against mouse IgG and rabbit IgG were obtained from Jackson ImmunoResearch Labs (West Grove, PA, USA) and probed at 1:10,000 in the blocking solution. The western blot was imaged using PXi Touch (Syngene, Frederick, MD, USA).

### 4.5. RT-qPCR

Total RNA was isolated with RNeasy mini for RT-qPCR from HUVEC grown in a 6-well plate. RT-qPCR was performed using 1-Step Quantitative Reverse Transcription PCR from 100 ng of RNA (Biorad, Hercules, CA, USA) following the manufacturer’s description. The relative mRNA level was measured with primers purchased against GLUT-1, VEGF, and HIF-1α (Sigma) compared to PPIA and β-actin simultaneously. The data was analyzed by CFX Manager (Biorad).

### 4.6. Experimental Animals

The mice were maintained in pathogen-free facilities in the National Cancer Institute (Frederick, MD). The study was approved by the NCI Animal Care and Use Committee (protocol numbers 17-009, 17-010, and 17-048, 11 January 2017), and in accordance with the Animal Research: Reporting of In Vivo Experiments (ARRIVE) guidelines. VE-Cadherin-cre/ERT2 and Vav1flox/flox mice were obtained from the Jackson Laboratory (Bar Harbor, ME, USA) and crossed to establish the colony and backcrossed to C57BL/6 mice. Age and sex-matched mice were used for experiments. Blood glucose was measured with blood drops from tail tips of mice using a glucometer (Roche, Basel, Switzerland).

### 4.7. Fear Conditioning

The conditioning apparatus consisted of a mouse shock chamber (Coulbourn Instruments, Allentown, PA, USA) set up in a sound-attenuated box. On day 1, the conditioning day, after a 2 min acclimation period to the conditioning chamber, 8-week old female mice received three conditioning trials consisting of a 30 s presentation of a 5 kHz, 70-dB tone [conditioning stimulus (CS)] that co-terminated with a 0.7 mA foot shock delivered through the grid floor during the last 1.0 s of the sound. Each conditioning trial was separated by a 30 s intertrial interval (ITI). After conditioning, mice were returned to their home cages. Mice were traced with infrared beam cue for subsequent quantification of behavior. Time spent “freezing” before and during the presentation of the CS tone was measured during the CS presentation as well as during a 30 s baseline period before the first tone trial. This latter measure served as an assay for unconditioned effects of the CS on general activity levels. Memory for the context and the tone was evaluated on days 2 and 3, respectively (~24 and 48 h after conditioning). For the contextual test, mice were placed in the conditioning chamber and allowed to explore for 2 min, after which freezing to the context was assessed for the remaining 4.5 min. For the tone test, mice were placed in a novel chamber with green walls. Mice could acclimate to the chamber for 2 min and be then presented with the CS (tone) on three consecutive trials (30 s, 5 kHz, 70 dB; ITI of 40 s). Freezing was evaluated during the 2 min acclimation period, during each presentation of the tone CS, and during the 40 s intertrial interval. After the memory tests, animals were returned to their home cage and colony. Memory for either the context or tone CS was quantified by the percentage of time engaged in a fear-related behavior (freezing) during context testing or CS presentation. Ten 4-OH tamoxifen or vehicle injected VE-Cadherin-cre/ERT2 and Vav1flox/flox mice were used for this experiment.

### 4.8. PET/CT Scanning

PET/CT scanning was performed after 8-week old female mice were fasted for 12 h but had free access to water. Following determination of body weights (28.0 ± 3 g on average), mice were anesthetized using vaporized isoflurane (4% for induction; 2.5% for maintenance). Sterile normal saline (0.1 mL) was injected subcutaneously to ensure adequate hydration. Following administration of ^18^F-FDG (<200 μCi) via lateral tail vein injection, mice were awakened and returned to individual holding cages. The optimal measurement time was determined after the kinetic study of the intravenous ^18^F-FDG injection. After the comparison of the dynamics of ^18^F-FDG signal from brain, heart and liver, 60 min post-injection was elected to compare the radioactivity from Vav1 deficient mice (Appendix A). Next, precisely 60 min post-FDG injection, anesthetized mice were placed prone in a heated (36 °C) multimodality chamber) in the PET scanner’s gantry. PET list-mode data were acquired for 15 min, using a small-animal PET/CT imaging system (GE Healthcare, Laurel, MD, USA). In the same workflow, a CT image was acquired for attenuation correction purposes. Images were reconstructed using a three-dimensional OP-MAP algorithm (Siemens Medical Solutions USA, Inc., Malvern, PA, USA). Scanner reconstructions using this algorithm were calibrated back to a National Institute of Standards and Technology (NIST)-traceable positron-emitting dose calibrator source to assure quantitative accuracy. Standard Uptake Value (SUV) was normalized by injection activity (µCi) and animal body weight (g). Twelve 4-OH tamoxifen or vehicle injected VE-Cadherin-cre/ERT2 and Vav1flox/flox mice were used for this experiment.

### 4.9. Histology

Brain, heart, skeletal muscle, and pancreas were obtained from both wild type and endothelial Vav1-deficient female mice that were eight weeks old after perfusion with PBS. For standard histology, formalin-fixed tissues were embedded in paraffin, sectioned in 5 μm, and stained (H&E) by the NCI-Frederick Histology Core.

### 4.10. Statistical Analysis

All statistical analyses were carried out using Prism 7 (La Jolla, CA, USA). Quantitative variables were analyzed by the paired Student’s t-test and two-way ANOVA test with multiple comparisons. All statistical analysis was two-sided, and *p* < 0.05 was considered statistically significant.

## Figures and Tables

**Figure 1 ijms-21-01341-f001:**
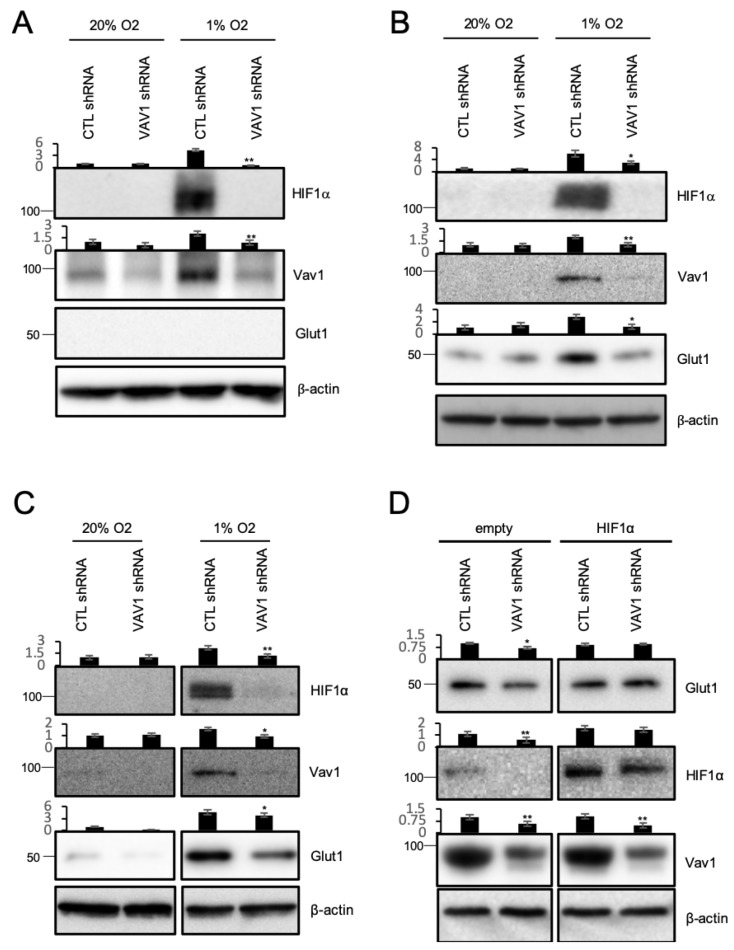
Vav1 regulates HIF-1α and GLUT-1 in endothelial cells. (**A**) HUVEC cells with control or Vav1 shRNA were incubated under normoxia or 1% O_2_ conditions for 5 h, or (**B**) 16 h. Cells were lysed to measure HIF-1α, Vav1, and GLUT-1 levels by Western blotting. (**C**) hCMEC/d3 cells with control or Vav1 shRNA were incubated under normoxia or 1% O_2_ conditions for 16 h. Cells were lysed to measure HIF-1α, Vav1, and GLUT-1 levels by Western blotting. (**D**) Control or Vav1 shRNA expressing hCMEC/d3 cells were transfected with an empty vector or HIF-1α overexpression vector. Cells were incubated under 1% O_2_ conditions for 16 h and then subjected to Western blot. Each experiment was repeated three times, and representative data is selected. (* *p* < 0.05, ** *p* < 0.01, mean ± SD).

**Figure 2 ijms-21-01341-f002:**
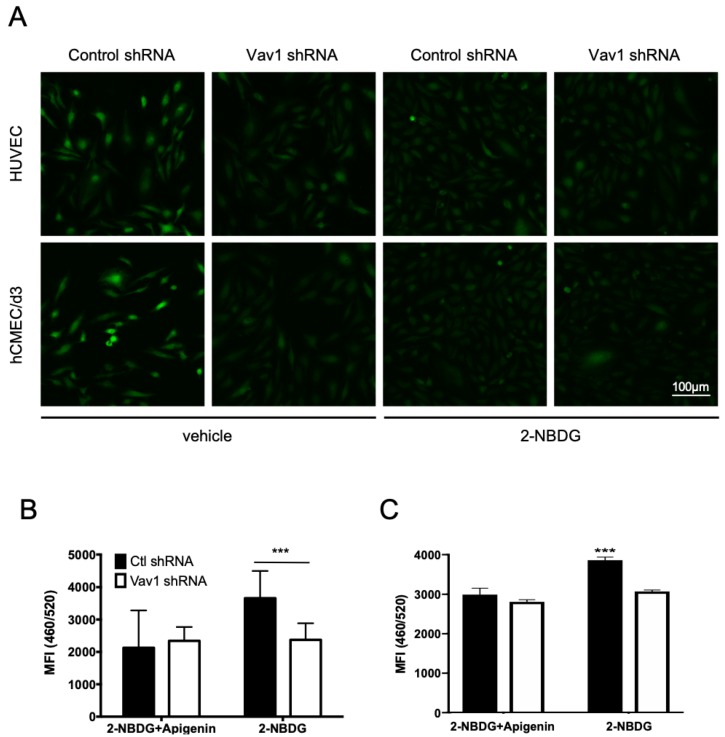
Vav1 regulates glucose uptake in endothelial cells. (**A**) HUVEC and hCMEC/d3 cells with control or Vav1 shRNA were starved in glucose-free medium for 1 h before the addition of 2-NBDG under hypoxic conditions with or without apigenin. After 1 h of incubation, cells were fixed with paraformaldehyde and observed by confocal microscopy. HUVEC (**B**) and hCMEC/d3 (**C**) cells with control or Vav1 shRNA in a 96-well plate were starved in glucose-free medium for 1 h and then treated with 2-NBDG with or without apigenin under hypoxic conditions for an hour. Mean fluorescence intensity (MFI) was measured to quantify 2-NBDG uptake levels (quadruplet mean ± SD). *** *p* < 0.001.

**Figure 3 ijms-21-01341-f003:**
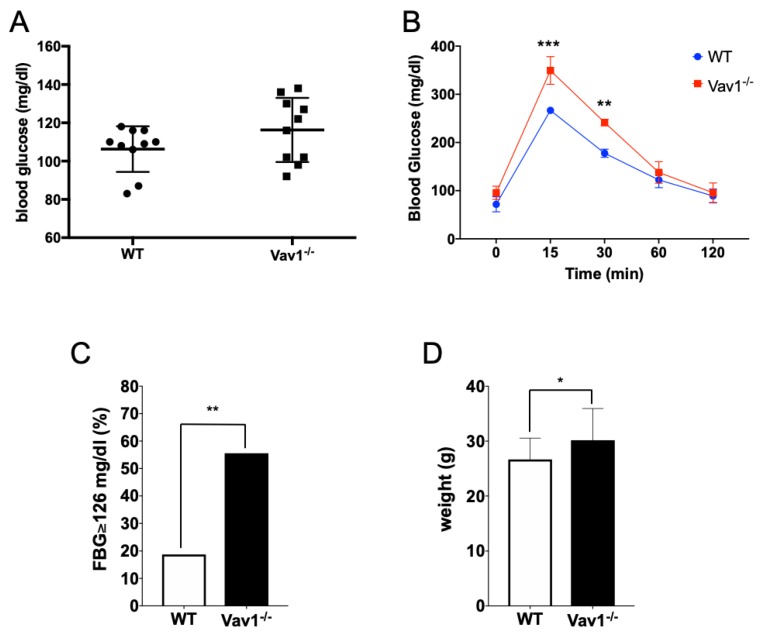
Vav1 deficiency delays glucose distribution in vivo. 8-week old female wild type or endothelial Vav1-deficient mice have fasted for 12 h in the presence of water. (**A**) Blood glucose levels were measured from the tail tip of each mouse (*n* = 10/group). (**B**) After 12 hours’ fasting, mice were given with 2 g/kg of glucose intraperitoneally. The blood glucose level was measured at each designated time point (*n* = 10/group). (**C**) The percentage of animals with 126 mg/dl or higher fasting blood glucose (FBG) in the basal state is displayed (*n* = 12/group). (**D**) The weight of mice measured after 12-h fasting (*n* = 12/group). Mean ± SD, * *p* < 0.05, ** *p* < 0.01, *** *p* < 0.001.

**Figure 4 ijms-21-01341-f004:**
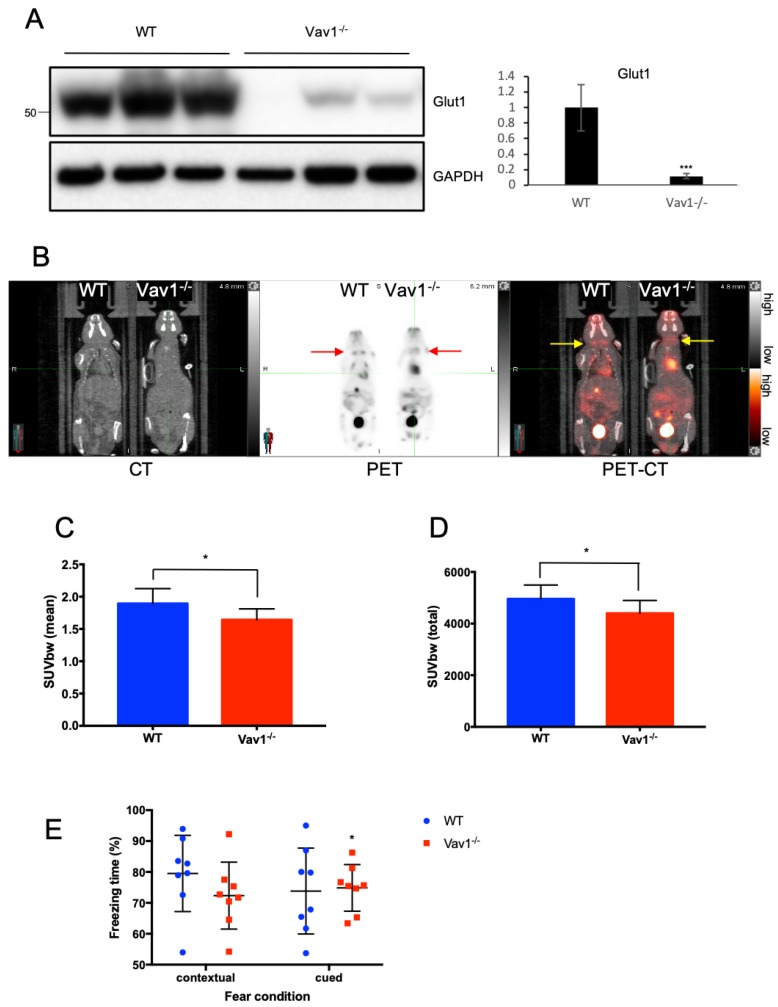
Vav1 deficiency downregulates glucose distribution in the brain. (**A**) The total lysate of mouse brain was subjected to Western blot to measure GLUT-1 protein expression levels (*n* = 3/group). Mean ± SD, *** *p* < 0.001. (**B**) Wild type or endothelial Vav1-deficient mice were subjected to a PET/CT scan to measure the distribution of 18F-FDG. Mice fasted for 12 h before intravenous injection of 18F-FDG (Arrows indicate brain hemispheres). One hour after 18F-FDG injection, the whole body of mice was scanned. The scanned signal in brains was normalized and then numerated as mean (**C**) and total (**D**) standardized uptake value body weight (SUVbw) (*n* = 10/group). (**E**) Wild type or endothelial Vav1-deficient mice were subjected to a fear conditioning test. Both contextual and cued conditions were given to measure the percentage of time in a frozen state (*n* = 10/group). Mean ± SD, * *p* < 0.05. All mice were female and eight weeks old.

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
