# Peer review of "The Rho/Rac Guanine Nucleotide Exchange Factor Vav1 Regulates Hif-1α and Glut-1 Expression and Glucose Uptake in the Brain"

_ijms, 2020, doi:10.3390/ijms21041341_

Round 1

Reviewer 1 Report

The authors evaluate the Rho/Rac guanine nucleotide exchange factor Vav1 in the regulation of Hif-1α and Glut-1 expression and glucose uptake in the brain of an animal model experiment. Their research is innovative and significant but they have failed to conform to the format of the proper reporting of their study. My one and only request would be to write their manuscript according to ARRIVE Statement. 

Author Response

Dear Reviewer,

We really appreciate your recognition of our novel finding.

According your request, we added animal experimental information including ACUC information.

We also added the number of animals used for each experiment in the manuscript.

Let us know if there is any necessary changes to be made.

Thanks,

Jaewoo Hong

Reviewer 2 Report

The authors investigated vav1 effects on glucose homeostasis under hypoxia. 

the topic is interest, but the results presented do not sufficiently support the study conclusions. 

The introduction is not supported by relevant literature.

It would be pertinent to describe the glucoregulation hormones of the transgenic mice. 

PET kinetic analysis is necessary to detail glucose delivery and consumption.

all results need appropriate statistical analysis to support their interpretation.

Author Response

Dear Reviewer,

Really appreciate your effort and time on our manuscript.

Your comments on our manuscript does not only revise the writing but also increase the quality of our manuscript remarkably.

All your comments were so suitable and reasonable, that we prepared point-by-point answer to each of your review.

-------------------------------------

The authors investigated vav1 effects on glucose homeostasis under hypoxia. 
=>This is exactly what we want to state through our writing, especially in the glucose homeostasis in the brain area.

the topic is interest, but the results presented do not sufficiently support the study conclusions. 
=>We noticed a lot of errors in our writing, so we corrected entirely.

The introduction is not supported by relevant literature.

It would be pertinent to describe the glucoregulation hormones of the transgenic mice. 

=> We added proper literatures as reference to explain the background of GLUT-1 biology, the character of Vav1 and our previous finding of the Vav1 mediated HIF1 control mechanism. The whole new introduction will give much better quality to our writing.

PET kinetic analysis is necessary to detail glucose delivery and consumption.

=> We added our kinetic analysis of 18F-FDG data to the supplementary data. We already finished the kinetic study before the comparison of several animals at 60 minute time points. Actually the 60 minute post-injection was determined after the kinetic study showing the whole body with radioactive signal.

all results need appropriate statistical analysis to support their interpretation.

=> We added Student's statistical analysis to each figure legend for the reliability of the data. We put the statistical analysis of Western Blot data too.

-------------------------------------------------------

We wish this answers your comments from our manuscript. We include the revised manuscript with figures and supplementary materials.

We wish you to have a happy new year.

Thanks,

Jaewoo

Round 2

Reviewer 2 Report

The revised manuscript would benefit from 2 minor alterations. The authors should mention the gender and age of the animals used in the study (animals' section). Given the low sample sizes across all figures, dot-plots should be preferred over bar-graphs.

Author Response

Dear reviewer,

We, authors, really appreciate your effort and time consumption in our manuscript. For this second revision round once more.

We definitely understand the points of your comments and I believe they are very beneficial.

Below are you will find the answers to your comments in this review round.

1. The authors should mention the gender and age of the animals used in the study (animals' section).

=>This was obviously our mistake not including the sex and age information of animals. We updated the materials and method section and figure legends to include this information

2. Given the low sample sizes across all figures, dot-plots should be preferred over bar-graphs.

=> Basically, >10 mice for each experimental group are not too small numbers to represent the population, weight information, so the bar graph is good enough to display the data in our belief. Additionally, SUV values are presented by two different methods in figure 4. Hence, the dot plot does not seem to be critically necessary for our data. Additionally, the data was statistically meaningful (P <0.05, minimum), so I wish you to understand this part.

I believe this may be an understandable answer to your comments and suggestions. Despite we do not correct every single point, we want this to convince your thought.

Thanks,

Jaewoo